# Chain Reaction of Behavioral Bias and Risky Investment Decision in Indonesian Nascent Investors

**Rika Dwi Ayu Parmitasari [1,*], Alim Syariati [1]** 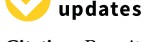 **and Sumarlin [2]**

1    Management Department, Universitas Islam Negeri Alauddin Makassar, Gowa 92118, Indonesia; alim.syariati@uin-alauddin.ac.id
2    Accounting Department, Universitas Islam Negeri Alauddin Makassar, Gowa 92118, Indonesia; sumarlin.habibi@uin-alauddin.ac.id
*    Correspondence: rparmitasari@uin-alauddin.ac.id

**Abstract:** Early investors possess unique sets of decision-making characteristics. They are more open to experience and eager to face risks. However, to the best of the authors' knowledge, the discussions of nascent investors upon making the investment decision and its eroding biases were still elusive. The vital role of emotion as a bias in decision making was also inadequately addressed. This study enhanced behavioral finance knowledge by examining emotion's role in regulating the illusion of control, overconfidence, and investors' decision making. In total, 456 initial investors in Indonesia participated in online questionnaires, forming the data for covariance-based structural model analysis. This study found that emotion significantly increased the illusion of control, but not overconfidence or decision making, contrary to the bulk of previous studies. The illusion of control exhibited a substantial significant effect of as much as 86.4% toward overconfidence, followed by a considerable increase in decision making. The results of our study also pointed to the unique chain effects of biases affecting the decision-making process of nascent investors in the emerging market. This finding implied they possessed a unique bias mechanism in constructing their decision.

**Keywords:** biases; decision making; the illusion of control; overconfidence; the nascent investors



## 1. Introduction

The discussion of decision making was a long-investigated topic in psychology, additionally, financial behavior (Kumar and Goyal 2015). The efficient market theory of Fama and expected utility theory described how investors decided their course of investment rationally based upon available information (Ackert et al. 2003); however, people are not always rational. They showed bias in the later stage of knowledge, despite initially being rational (Kumar and Goyal 2016; Guo et al. 2022), simply because the presence of risks in investment could point to the bounded rationality of investors (Kahneman and Tversky 1979; Tobin 1958; Yao and Li 2013; Simon 2000). Furthermore, age composition revealed a different set of confidence in the effectiveness of decision making, as older adults rated their problem solving better but lack the determination relative to youth or middle-aged people (Thornton and Dumke 2005; Bouteska and Regaieg 2018; Putu et al. 2022; Prihatini et al. 2022). The high emotional salience condition makes the young and middle-aged more proactive in regulating emotion than passive older adults (Blanchard-Fields et al. 2004; Yeung et al. 2012). In gambling, young adults were more avoidant of negative feedback than other age demographics (Cauffman et al. 2010) and promoted risky behavior related to reward aspiration (Van Leijenhorst et al. 2010).

Compared to experienced investors, young investors have a different composition of behavioral sets in investment decisions. Greenwood and Nagel (2009) pointed to their tendency to follow trends, thus creating a stock price bubble and heavily investing in technological firms. A study in India pointed toward early investors' spending more than half of their income in the stock market while balancing the income/growth factor and its

risk (Ansari and Moid 2013). Risk-averse characteristics are also present in their features (Wood and Zaichkowsky 2004). They were also eager to discuss their problem with online friends or in the school environment (Tan and Tan 2012; Shim et al. 2009), although other studies pointed out the importance of parents in shaping the attitudes toward finance (Shim et al. 2015). Other academics such as Dimov (2010) discussed it in the context of nascent entrepreneurs, defined as those starting their entrepreneurial endeavors. He demonstrated cyclical confidence in investing, given the feasibility of the business prospects and knowledge in possession. He also showed that their opportunity confidence and industry experience stopped them from discontinuing their business aspiration. Nascent entrepreneurs with advanced education were eager to engage in the market and were ready to jump into action, but not on the planning (Ko and Wiklund 2016). That study exhibited the signs of creating a bias in decision making, as they left the rational process of decision making behind in favor of hasty reactions. However, as they accumulated more experience and knowledge, they would proceed better in later life (Zacharakis and Shepherd 2001; Fabricius and Büttgen 2013; Gort et al. 2008; Li et al. 2019; Misuraca et al. 2022).

Apart from those characteristics, previous studies rarely discussed the psychological bias affecting nascent investors' decision making. Previous studies mainly probed into a general audience of investors. Examining this demographic uniqueness would contribute to the minor discussions in the field. Another contribution came from the analysis of emotion as another bias in pushing risky decision making, as it is not as much investigated as other biases, e.g., overconfidence, self-attribution bias, and so forth (Singh and Bhattacharjee 2019). This study statistically tested emotion as a driver of the illusion of control, overconfidence, and decision making in a direct and indirect relationship. We found a unique finding: the links formed a chain reaction from a small effect of emotion to the illusion of control, which substantially increased the overconfidence, followed by the decision making. Several studies grouped the illusion of control as another function of overconfidence (Bhandari and Deaves 2006; Chu et al. 2012; Johnson and Fowler 2011; Lambert et al. 2012), to which the authors agreed, considering the substantial effect between the two constructs. This study was critical in opening the black box of the young-investor decision-making process on the associated biases. Selected demographic and geographic settings also bestowed further explanations.

*Hypotheses*

In the efficient market theory, a market with information available to all investors would render them rational. However, financial behavior emerged because investors were sometimes ir-rational (Kumar and Goyal 2015). Kahneman and Tversky (Kahneman and Tversky 1979; Sarapultsev and Sarapultsev 2014), with their prospect theory, explained that the outcomes were not the only considerations of investors, but a variety of potential gains and losses also penetrated their minds. The interaction of return and risk could make investors ir-rational. This ir-rational behavior raised several real impacts, one of which is the emergence of bias in decision making (Kumar and Goyal 2016; Ricciardi 2008). Some preferences were able to influence decisions, e.g., overconfidence, disposition effect, herding effect, mental accounting, confirmation bias, hindsight bias, house money effect, endowment effect, loss aversion, framing, home bias, self-attribution bias, conservatism bias, regret aversion, recency, anchoring, and representativeness (Zahera and Bansal 2018). This study examined the impact of emotion, illusion of control, and overconfidence toward early investor decision making, in direct or indirect relationships.

Affection is a part of bounded rationality in decision making (Clark 2010; Muramatsu and Hanoch 2005). Emotions have a variety of actual effects on humans. In investment activities, the negative role of emotions has been well documented as one of the biases of decision making (Seo and Barrett 2007), reflected in stock prices (Kuzmina 2010). Seo and Barrett's study indicated three facts of emotion as a bias. They are: (1) emotions affected the quality of information absorption in the brain; (2) the role of emotions in the creation of judgment on a condition is widely documented; (3) the actual impact on

decision making. The experiment found that emotions promoted risky decisions, but if they emerged naturally, they could increase risk-averse conditions (Heilman et al. 2010). Investors who can control emotions have better investment returns than reactive ones (Fenton-O'Creevy et al. 2011).

Reasonable emotional control has an impact on better decision making. Emotions could drive positive investor outcomes in the financial markets (Ackert et al. 2003). Experimental results also confirmed a positive relationship between trait anxiety and gambler performance and even showed that high emotional control would reduce gambler's outcomes (Werner et al. 2009). Extensive experiments in patients with a lesion in the region of the brain that controls emotions produced more favorable results than healthy patients (Shiv et al. 2005a, 2005b). Affect also created the illusion of control (Alloy et al. 1981; Alloy and Abramson 1982; Mesken et al. 2005). Emotions were also an essential factor that could increase overconfidence (Chu et al. 2012; Köther et al. 2018; Treffers and Fehse 2016); thus:

**Hypothesis 1.** *Emotion was one of the foundations of the illusion of control.*

**Hypothesis 2.** *Emotion was the positive driver of overconfidence in early investors.*

**Hypothesis 3.** *Emotion also regulated the decision making of nascent investors.*

The illusion of control in Langer's definition is the exaggerated expectation of the possibility of becoming successful compared to the actual result. Another simple explanation is called unrealistic optimism (Harris and Middleton 1994). However, McKenna clearly distinguished unrealistic optimism and illusion of control, where optimism was optimistic hopes independent of the origin. In contrast, the illusion of control determined that the source of that expectation is in one's power and dominates decision making compared to optimism (McKenna 1993). As a bias, the remedy to the illusion of control could emerge from external advice to restore the individual's rationality (Meissner and Wulf 2016). As a bias, the illusion of control had some relationships with other constructs.

Replicating the work of Langer, the experiments showed that the absence of the illusion of control was the initial sign of feelings of incompetence in oneself (Golin et al. 1979; Langer 1975). Other studies documented a link between locus of control and trust in superstitious activities (Rudski 2004; Haerani et al. 2019; Parmitasari et al. 2018). Recent discussions had taken further directions. Those with a high level of locus of control exhibited some distinguishing characteristics. They were more active in job-seeking (Caliendo et al. 2015; McGee 2015), they migrated more frequently (Caliendo et al. 2019), or they saved more money (Cobb-Clark et al. 2016). Gamblers also display a high tendency toward the illusion of control, as is the nature of gambling itself. They used their past highest win as a measure of future income potential for those with a high illusion of control. At the same time, the lower ones focused their attention on the outcome of their action (Cowley et al. 2015). The focus on ultimate victory stemmed from the inability to discriminate between good and evil; furthermore, it depended on the severity of the gambling symptom (Perales et al. 2017). The meta-analysis showed that motivation was more significant than skill in encouraging the illusion of control (Stefan and David 2013). Brain imaging revealed the absence of consistency between those who experienced the illusion of control or not (Kool et al. 2013). Other studies also proved that the illusion of control was not the predictor of the winning convincement (Filippin and Crosetto 2016). These findings indicated that there was still no consistency in discussing the illusion of control (Stefan and David 2013). This research analyzed the relationship between the illusion of control and overconfidence.

As a bias, no consensus emerged from the relationship between the illusion of control and overconfidence. Some studies included the illusion of control as an independent construct toward overconfidence (Fortune and Goodie 2012; Kannadhasan et al. 2014; Simon et al. 2000). Other academics argued that the illusion of control was a dimension that formed overconfidence (Bhandari and Deaves 2006; Chu et al. 2012; Johnson and Fowler 2011; Lambert et al. 2012). These differences resulted in inconsistencies in constructing

relationships. This study classified the illusion of control as a construct that influenced overconfidence in a risk framework (Lam and Ozorio 2014; Hilton et al. 2011). Based on the above theoretical discussion, we have:

**Hypothesis 4.** *Inceptive investors with a high illusion of control exhibited high overconfidence.*

Overconfidence has been the subject of investigation in many studies (Azam et al. 2022). In the context of investment, overconfidence is an excessive belief in knowledge and skills in executing the investments (Kumar and Goyal 2015; Zahera and Bansal 2018). Moore and Healy (2008), in their literature review, grouped a variety of definitions of overconfidence, which come on three notions, (1) overestimation, overprediction of individual ability; (2) overplacement, mis-specification of the outcome compared to others, and (3) overprecision, a strong favor and belief toward one's action. Their investigation revealed that 64% of the studies on overconfidence use the first definition, 5% use the second, and 31% use the third. They found that many studies treated those three definitions as the same. To reconcile those inconsistencies, they then proposed the theory that investors' overconfidence results from the lack of performance information in post-tasks and even worse details of others. Other studies amplified Moore and Healy's finding by proving that the increased information would contribute more to overconfidence (Fabricius and Büttgen 2013; Gort et al. 2008). Moore and Healy (2008) also revealed the third definition of overconfidence as the more persistent notion, even reducing the presence of overestimation and overplacement. The underlying conditions played a critical role in administrating those discrepancies (Li et al. 2017). Overconfidence is also one of the most frequent measures of bias in decision making.

This study examined the relationship between overconfidence and nascent investor decision making. The negative consequence of overconfidence was the reduced accuracy of decision making (Zacharakis and Shepherd 2001). Overconfidence investors exhibited a higher risk tolerance (Heilman et al. 2010; Pikulina et al. 2017). They regularly executed transactions (Glaser and Weber 2007; Malmendier and Tate 2005, 2015). They also favored short-term debt (Huang et al. 2016). At the CEO level, their confidence often causes stock crashes or execution acquisitions, in line with their domination in the organization (Al Mamun et al. 2020; Brown and Sarma 2007; Kim et al. 2016). There were several contexts in the relationship between overconfidence and decision making. The evidence in India revealed that men are more prone to overconfidence (Kumar and Goyal 2016). Extensive knowledge acquisition also boosted investor overconfidence (Fabricius and Büttgen 2013; Gort et al. 2008). Nascent investors also demonstrated high overconfidence during their transactions (Gervais and Odean 2001). Based on all previous studies, we have:

**Hypothesis 5.** *Overconfident nascent investors would make more investment decisions.*

**Hypothesis 6.** *Emotion would increase the overconfidence through the illusion of control.*

**Hypothesis 7.** *Emotion would indirectly improve investors' decision making following the path of the illusion of control and overconfidence.*

**Hypothesis 8.** *Illusion of control would increase the overconfidence and subsequently the decision making.*

## 2. Materials and Methods

### 2.1. Sample

This study collected data from the nascent investors who are becoming members of the Indonesian Stock Exchange Investment Gallery (IDX Investment Gallery). This Gallery is an effort by the Government of Indonesia and the Indonesian Stock Exchange to introduce security investments to academics, especially early undergraduate students. This study gathered data from young investors from four contributing universities in Indonesia.

We managed to code the data from 456 students. In total, 59.76% of them were female. Despite being undergraduate students, 10 of them, or 3.9%, were already married. Many of the participants were business students, who constituted the majority of 83.9%, and others were language, engineering, education, and law students. This study employed several enumerators to make a face-to-face meeting, resulting in a 100% return rate. We also found no missing data in the response, as we used the google form to ensure the accuracy of answers.

*2.2. Measures*

This study employed a 5-point Likert scale depicting "strongly disagree" (1) and "strongly agree" (5) to each questionnaire item. Aside from demographic factors, i.e., ages, marital status, and educational status, this paper collected information from 28 questionnaire items. Those items represented the latent variables of emotion, the illusion of control, overconfidence, and the decision making of young investors. We extracted 28 indicators reflecting those latent variables from several studies and translated them into Indonesian to avoid confusion. Initially, we conducted a pilot study to check the items on 30 people to reach more valid and reliable signs (Hair et al. 2010). We rewrote or deleted the low-score items to obtain better responses using confirmatory factor analysis (CFA) by using the Lisrel 10 software. This approach is critical in the covariance-based structural-equation modeling to check whether the indicators can sufficiently represent the said variables. The structural equation modeling may help assess an abstract latent construct by its manifest scales in the survey setting. A total of 28 initial indicators were reduced into 13 scales reflecting four constructs, as in Table 1. We then checked the validity and reliability of the model by the convergent validity, discriminant validity, and collinearity test as recommended by Hair et al. (2010) by using the Smartpls 3 software. The convergent validity is measured by the outer loading of the indicators and the average variance extractor (AVE). The rho_a and composite reliability (CR) would ensure that the model demonstrates that the indicators have good reliability. Lastly, the variance inflation factor (VIF) indicates that the model is absent from the multicollinearity problems. These data-quality measures are the prerequisite criteria before constructing the path model for hypothesis testing.

**Table 1.** Research indicators.

| Latent Constructs | Indicators | Items |
|---|---|---|
| Emotions | I panic during the market breakdown | EMO1 |
| | I am happy to invest more when profiting | EMO2 |
| | I hastily buy during a bullish market | EMO3 |
| Illusion of Control | I always know where to invest | ILC1 |
| | The market risks are always within my perimeter | ILC2 |
| | The return is always within my perimeter | ILC3 |
| Overconfidence | I am a good investor | OVC1 |
| | My investment analysis is better than my fellow friends | OVC2 |
| | My investment strategy works perfectly in most cases | OVC3 |
| Risky Decision Making | I directly cut loss when bearish | IDE1 |
| | I directly sell when the market is bearish | IDE2 |
| | I always follow the highest potential return in investing | IDE3 |
| | An opportunity is mostly a single shot | IDE4 |

This study developed five statements to measure the illusion of control. This scale development would follow the proposition that investors would be different in selecting several investments using their judgment or following the advice of gallery supervisors

or professional investors (Meissner and Wulf 2016). They pointed to the tendency toward self-selecting the investment portfolios as a form of the illusion of control. The indicators reflected the ability of nascent investors to select potential investments, reduce the upcoming risks, believe in the easiness of investing, and predict the results of investments ($\alpha = 0.650$). To measure overconfidence, we used five items representing self-attribution, the better-than-average effect, and the planning fallacy of investors ($\alpha = 0.472$) (Kansal and Singh 2018). We assessed early investors' primary and secondary emotions using seven statement scales (Kuzmina 2010). The primary emotions reflected happiness, sadness, anger, or fear; on the other hand, secondary emotions formed from euphoria, melancholy, or even panic ($\alpha = 0.696$). Finally, we expected that young investors were rational upon making a decision and thus developed ten items of investors' decision making that express dependency, rationality, and intuitiveness ($\alpha = 0.690$). The scales in our study are in Table 1.

Using questionnaires at one point could result in common method bias (Podsakoff et al. 2003). Thus, this study employed Harman's single factor test to observe the potentiality of this problem by using SPSS. This study found that all indicators' variance is 20.185%, indicating the nonbias in this study. Table 2 exhibited the means, standard deviations, and correlations of the model.

**Table 2.** Mean, standard deviation, and correlations.

|   | Constructs | Mean | St. Dev. | 1 | 2 | 3 | 4 |
|---|---|---|---|---|---|---|---|
| 1 | EMO | 2.86 | 0.84 | 1 | | | |
| 2 | ILC | 4.05 | 0.62 | 0.158 * | 1 | | |
| 3 | OVC | 3.59 | 0.62 | 0.168 ** | 0.502 ** | 1 | |
| 4 | IDE | 3.96 | 0.59 | 0.193 ** | 0.409 ** | 0.350 ** | 1 |

\* $p < 0.05$; \*\* $p < 0.01$.

## 3. Results

The initial assessment was performed to check the validity and reliability of the data. We tested composite reliability, discriminant validity, and discriminant validity using SmartPLS 3. These tests were recommended by Hair et al. (2010) when a researcher is executing the structural equation modeling to represent the quality assessment of the model, as evident in Table 3.

**Table 3.** Validity and reliability measures.

| Construct | Items | VIF. | Loading | $\alpha$ | Rho_a | CR | AVE |
|---|---|---|---|---|---|---|---|
| Emotion | EMO1 | 1.360 | 0.832 | 0.696 | 0.750 | 0.819 | 0.605 |
| | EMO2 | 1.323 | 0.634 | | | | |
| | EMO3 | 1.385 | 0.849 | | | | |
| Illusion of control | ILC1 | 1.133 | 0.622 | 0.650 | 0.683 | 0.811 | 0.592 |
| | ILC2 | 1.475 | 0.842 | | | | |
| | ILC3 | 1.458 | 0.825 | | | | |
| Overconfidence | OVC1 | 1.109 | 0.768 | 0.472 | 0.491 | 0.736 | 0.485 |
| | OVC2 | 1.071 | 0.571 | | | | |
| | OVC3 | 1.166 | 0.735 | | | | |
| Risky decision making | IDE1 | 1.462 | 0.770 | 0.690 | 0.716 | 0.812 | 0.524 |
| | IDE2 | 1.132 | 0.555 | | | | |
| | IDE3 | 1.378 | 0.701 | | | | |
| | IDE4 | 1.839 | 0.838 | | | | |

Source: Smartpls 3 output.

The initial validity and reliability tests exploited the Smartpls 3 tool to communicate the quality criteria of the model. The initial internal consistency measured from the Cronbach's alpha, rho_a, and the composite reliability (CR) revealed whether the observed

constructs were consistently reliable in the model running with the cut-off value higher than 0.6 (Hair et al. 2014). Table 2 shows that all variables meet the standards, while overconfidence emerges as the construct with the lowest scores. However, as the CR was still higher than 0.7, we considered this variable's reliability to persist. The convergent validity measurement provides information on the indicators' loading factors and the average variance extractor (AVE). The statistical tests provided evidence that all indicators are higher than the 0.5, as Hair et al. (2014) recommended not to delete the items. The test indicated the AVE of all constructs to be higher than 0.5 aside from the overconfidence. However, Hair et al. (2014) did not recommend deleting an item or variable if such changes do not improve the model quality. This study also presented the test of collinearity statistics by using the variance inflation factor (VIF). Each item's VIF ranged from as low as 1.071 to as high as 1.839, indicating the absence of multicollinearity and the common method bias (Kock 2015, 2017). This study also reported the discriminant validity of the model by the Fornell–Larcker criteria as in Table 4.

**Table 4.** The Fornell–Larcker Test.

| Construct | Emotion | Illusion of Control | Investment Decision | Overconfidence |
|---|---|---|---|---|
| Emotion | 0.778 | | | |
| Illusion of control | 0.176 | 0.770 | | |
| Investment Decision | 0.183 | 0.426 | 0.724 | |
| Overconfidence | 0.181 | 0.524 | 0.364 | 0.697 |

Source: Adapted Smartpls 3 output.

This study conducted the discriminant validity to assess whether one construct was empirically distinct from other unrepresented variables. The statistical evidence proved that all latent variables were correlated within constructs with higher scores than nonmeasured ones. This finding constructed the path for another quality test.

This study used Lisrel 10.20 to analyze the data, using either Confirmatory Factor Analysis (CFA) or the structural model (Amar et al. 2021). As most of the items were self-developed, we reanalyzed it through CFA and found the initial model to be less-fit ($\chi^2$ = 1122,49; $df$ = 344; CFI = 0.654; RMSEA = 0.0940). This finding convinced us to improve the model's fitness by deleting some items with low loading and high standardized residuals. This effort resulted in the decrease in items from 28 to 13 items only and found a better goodness of fit index ($\chi^2$ = 103.02; $df$ = 68; CFI = 0.959; RMSEA = 0.0449; GFI = 0.945). The high chi-squared score was predictable, as large sample size may erode the quality of that measure (Hu and Bentler 1999). Other criteria were still good to use, considering that the other absolute indexes such as RMSEA and GFI still fit (Hair et al. 2010). This second CFA indicated a good fit model and became the final data constructed in the path analysis in Table 5.

**Table 5.** Summary of Path Coefficients.

| | Path | Effect | T-Value | Hypothesis |
|---|---|---|---|---|
| H1 | Emotion –> Illusion of control | 0.204 | 2.179 * | accepted |
| H2 | Emotion –> Overconfidence | 0.104 | 1.095 | rejected |
| H3 | Emotion –> Investor decision making | 0.027 | 0.278 | rejected |
| H4 | Illusion of control –> Overconfidence | 0.864 | 5.026 ** | accepted |
| H5 | Overconfidence –> Investor decision making | 0.687 | 5.555 ** | accepted |
| H6 | Indirect emotion –> Overconfidence | 0.176 | 2.210 * | accepted |
| H7 | Indirect emotion –> Investor decision making | 0.192 | 2.450 * | accepted |
| H8 | Indirect illusion of control –> Investor decision making | 0.593 | 4.557 ** | accepted |
| | $R^2$ Illusion | | 0.042 | |
| | $R^2$ Overconfidence | | 0.813 | |
| | $R^2$ Investor decision making | | 0.379 | |

\* $p < 0.05$; ** $p < 0.01$.

The statistical analysis revealed that the early investors' emotion was predicting their illusion of control, but not their overconfidence or decision making. This result accepted Hypothesis 1 but not 2 or 3. Early investors with a strong illusion of control would be more overconfident in making an investment decision, confirming Hypotheses 4–8. The indirect relationship in Hypotheses 6–8 revealed a path mechanism where emotion could increase overconfidence or decision making by increasing the illusion of control first, instead of direct transmission. Regarding effect size, the relationship of the illusion of control toward overconfidence came with an enormous relevance of 86.4%. It was followed by overconfidence in investor decision making at 68.7%.

The illusion of control strongly predicted how participants made decisions indirectly, by as much as 59.3%. Other direct and indirect effects were relatively small, but they were acceptable, as this study was psychologically based. Funder and Ozer even argued that an effect size larger than 0.10 is small but still had some potential, while the value above 0.20 is medium, and 0.30 is large and substantial (Funder and Ozer 2019). In retrospect, some sizes were relatively small according to the commonly used power analysis (Cohen 1977). However, the nature of the behavioral-psychological study allowed for a smaller effect size than the investigation in organizational performance (Hair et al. 2014). Those studies confirmed that the size difference presented by our research was still acceptable. A better presentation of the path of revelation can be observed in Figure 1.

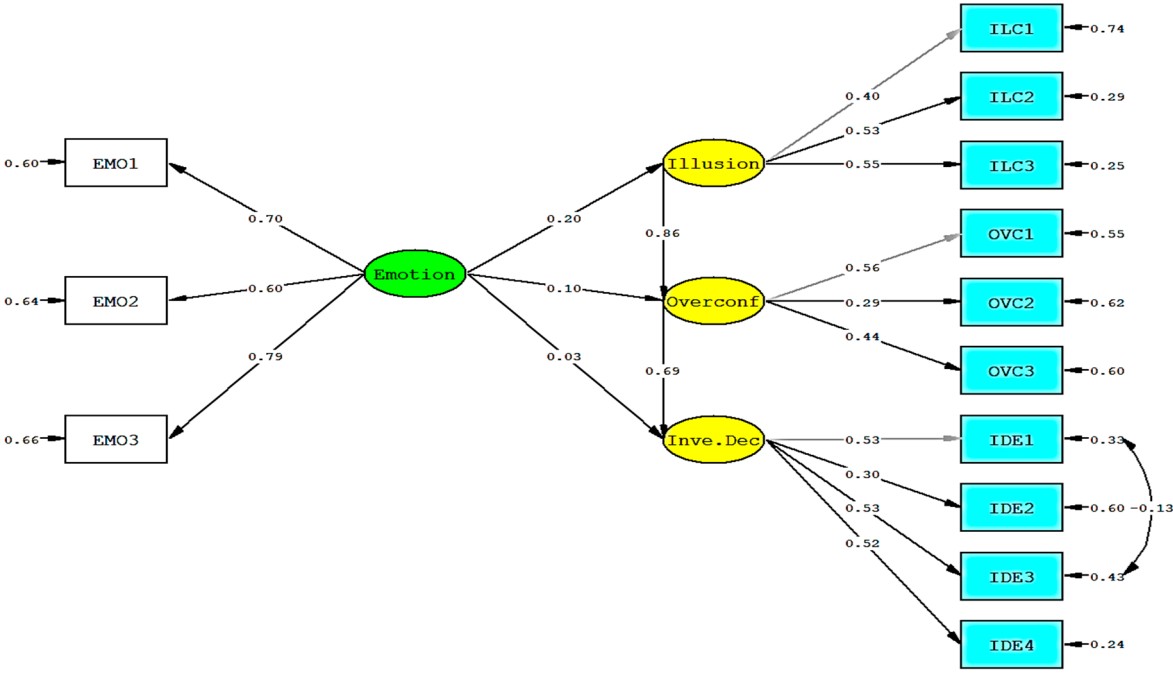

Chi-Square=84.27, df=59, P-value=0.01709, RMSEA=0.041

**Figure 1.** The Path Model (Source: Lisrel output).

## 4. Discussion

Experience in life and past or future expectations contributed to how one formed a decision (Juliusson et al. 2005). Even assuming humans are rational, decision-making bias will always be present. This study aimed to investigate some biases experienced by investors in making investment actions, especially those by early investors. This study also presented the findings in the context of a developing region, adding more evidence to the field.

This study found that emotions are one of the significant factors forming the illusion of control, confirming Hypothesis 1. A medium magnitude of 20.4% was seen in effect size (Funder and Ozer 2019). The role of emotions in forming the illusion of control was not conclusive yet. In transportation, experiments show that emotion is not a factor creating the illusion of control (Mesken et al. 2005). That experiment contradicted previous studies, which found that depressed individuals will make wrong decisions because of the emerging feelings of being able to control a situation, even when impossible (Alloy et al. 1981; Alloy and Abramson 1982). Other emotions, such as anger, also encourage a sense of being able to govern outcomes (Lerner et al. 2015; Lerner and Keltner 2001). People with emotional problems also make bolder decisions and obtain more significant potential gains or losses than ordinary people (Shiv et al. 2005b).

This research provided a unique contribution, namely, that nascent investors with adverse emotional ownership will encourage a feeling of being able to control something, even though it is beyond their power. Not surprisingly, they tend to be more reckless. However, different results are found on the emotional impact on overconfidence or decision making. This study discovered that emotion was not a driver of overconfidence or decision making in a direct relationship, thus rejecting Hypotheses 2 and 3. This result is a new finding, given that the majority of previous studies found emotions as one of the overconfidence and decision-making causes. Treffers and Fehse (2016) conducted unique research employing neuroimaging and found a transitory path from emotion to overconfidence leading to decision making from the cingulate cortex and hippocampus of the brain. Their findings also indicate that people who are not overconfident also have better decision-making performance (Treffers and Fehse 2016).

Medical studies show that overconfidence is a strategy to reduce ambiguity in decision making due to stressful pressures (Köther et al. 2018). In the financial sector, some emotions such as pride and shame are traits that arise when investors show symptoms of overconfidence, furthermore making them increasingly enlarge their investment portion (Chu et al. 2012). Overconfidence could be good or bad. Studies have indicated that managers with overconfidence had a slightly better financial performance (Harrison and Wicks 2013) and were more open to sharing their internal data (Libby and Rennekamp 2012). Another study found a contradicting result as overconfident managers had the worst financial performance during financial acquisitions (Doukas and Petmezas 2007). There is a strong relationship direction in indirect relationships, which proves that the emotions of early investors will develop toward the creation of the illusion of control, which triggers overconfidence and leads to decision making. Unlike other studies, our statistical revelation points to the absence of emotional relationships toward decision making.

The emotion of young investors did not directly encourage them to make investment-related decisions, rejecting the 3rd hypothesis. These findings contradicted most studies pointing to the critical role of emotions in decision making (Lerner et al. 2015). Several gambling studies also supported emotions' role in decision making (Fenton-O'Creevy et al. 2011; Werner et al. 2009). The trials on novice entrepreneurs also show the role of intuition rather than deep thinking when investing in unknowable situations (Huang and Pearce 2015). This intuition could further increase the use of emotion upon making critical decisions. This absence of relationship could come from several explanations.

Some experiments showed that cognitive reappraisal factors could reduce the negative impact of emotions on risky decision making (Heilman et al. 2010; Preston et al. 2007). Other studies also discovered that past failure might reduce painful emotions involved in investing (Shiv et al. 2005b). This finding could explain this study's lack of emotional connection and investment decision making. Nascent investors with limited financial capacity would be carefully concerned with each investment action. With a moment of introspection, they will further reduce emotions in critical decision making, escorting them to a more rational stance. Unlike feeling, the illusion of control was a strong predictor of overconfidence.

The illusion of control strongly and significantly influenced overconfidence, as observed from the t-value. These findings confirmed the 4th hypothesis. The high t-value, of more than 80%, is relatively rare in the context of behavioral science (Funder and Ozer 2019). Since its seminal appearance in 1975 (Langer 1975), the illusion of control has received many investigations. The relationship between the illusion of control and overconfidence also overlapped (Hilton et al. 2011). Several studies identified the illusion of control as a construct dependent on overconfidence, with an essential role in risk perception when starting a business (Fortune and Goodie 2012; Kannadhasan et al. 2014; Simon et al. 2000). However, other academics argued that the illusion of control is another strain of overconfidence (Bhandari and Deaves 2006; Chu et al. 2012; Johnson and Fowler 2011; Lambert et al. 2012). The findings of this study supported the role of the illusion of control as one of the most influential factors causing overconfidence, based on the magnitude of the relationship. Other research pointed out that social factors became the antecedent of overconfidence (Burks et al. 2013). A study showed that novice traders tend to be overconfident and, over time, become wiser (Gervais and Odean 2001). That study supported this study's findings regarding the substantial role of an illusion of control in increasing investors' overconfidence in the development region.

Finally, this study found that overconfidence positively and significantly affected investment decisions, endorsing hypothesis 5. Overconfident investors tended to display optimistic views during their early exposure to the trade (Gervais and Odean 2001) and were willing to absorb higher risks in making investments (Heilman et al. 2010; Pikulina et al. 2017). Zacharakis and Shepherd (2001) found that venture capitalists were people who were very overconfident in making decisions, dared to absorb more significant risks, and decreased the accuracy of decisions. This confidence also increased with information or knowledge

deepening and potentially reduced the quality of decision making. A study of a newsvendor competition revealed that overconfidence was good when profits increased and vice versa in a declining return, depending on certain circumstances (Li et al. 2017). The broader the investor's knowledge, the better the results of decision making for overconfident individuals would be (Fabricius and Büttgen 2013; Gort et al. 2008).

This study demonstrated exciting findings in the context of nascent investors in developing countries. Emotion was one of the factors forming the illusion of control but not directly related to overconfidence or decision making. Emotions formed a chain of indirect links between constructs, indicating a gradual change in behavior. The illusion of control was the most compelling and relevant factor creating overconfidence. The chain reaction potentially points to an increase in overconfidence upon making strategic investment decisions.

## 5. Conclusions

Economic actors would act in a manner that would best serve their interests. While they were built as rational creatures, they are bounded to potential biases in their decision making. This study constructs a model to probe this phenomenon in young investors in Indonesia. We capture the finding that early-aged traders build their roller-coaster of emotion during investment, further triggering their illusion of control and overconfidence. These compounding biases lead to the tendency to be more active upon executing an investment, even risky ones. Otherwise, emotion is not directly related to overconfidence and risky decision making.

This paper was still preliminary in discussing the behavior of young investors in the context of developing country with various weaknesses. Larger samples would provide better justification for the topic. Subsequent improvements could arrive from improved research design items and different demographic contexts. Further studies could also investigate different levels of knowledge in developing and developed countries making strategic decisions.

**Author Contributions:** Conceptualization, R.D.A.P. and A.S.; methodology, A.S.; software, A.S.; validation, R.D.A.P.; formal analysis, R.D.A.P.; investigation, R.D.A.P.; resources, R.D.A.P.; data curation, S.; writing—original draft preparation, R.D.A.P.; writing—review and editing, R.D.A.P. and A.S.; visualization, S.; supervision, R.D.A.P.; project administration, A.S. and R.D.A.P. All authors have read and agreed to the published version of the manuscript.

**Funding:** This research received no external funding.

**Institutional Review Board Statement:** Not applicable.

**Informed Consent Statement:** Not applicable.

**Data Availability Statement:** The study did not report any data.

**Conflicts of Interest:** The authors declare no conflict of interest.

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
