# Peer review of "Chain Reaction of Behavioral Bias and Risky Investment Decision in Indonesian Nascent Investors"

_risks, doi:10.3390/risks10080145_

Round 1

Reviewer 1 Report

The subject of the work, reaction of behavioral bias and risky Investment is potentially very interesting for readers. The presented analysis is very interesting. The paper is technically and theoretically sound and well-organized, although in my opinion "Introduction" should be consolidated and shortened, but "Conclusion" needs extension. For editional point of view, graphs are bad quality.

Author Response

Authors' Response: We thank you for your valuable comments. The MDPI journal employs IMRAD in the manuscript drafting, thus, making our introduction section seems bulky. Therefore, we deleted the introduction's theoretical section to shorten our paper slightly. We leave three paragraphs in the introduction and several subsections for the hypothesis development. We also have added a paragraph to extend our conclusion. The graph has also been changed to the one with a better pixel rate.

We wish these changes could meet your stated comments. We wish you good health.

Kind regards,

Authors

Reviewer 2 Report

1. Please briefly describe confirmatory factor analysis and structure equation models in Section 2.2.

2. In Table 3, what do Alpha, Rho_a, and CR stand for and their meanings? 

Author Response

Authors' Response: We thank you for your valuable comments. We have:

  1. Added several lines to provide a brief introduction of the quality assessment criteria in the confirmatory factor analysis process and the structural model.
  2. Clarified the alpha, rho_a, and CR by giving short explanation in the page 5, section 2.2.

We wish these changes could meet your stated comments. We wish you good health.

Kind regards,

Authors

Reviewer 3 Report

The study tries to delve into a field of research that joins the traditional principles of the theory or rationality of investors and the biases that behavioral economics and psycology have already revealed decades ago. 

That's the reason I apreciate the contribution of the paper.

A greater and deeper explanation of the results and their robustness analysis would permit better conclusions, beyond what the literature says in this regard

Author Response

Authors' Response: We thank you for your valuable comments. We have rewritten our result section to provide step-to-step of quality tests as on page 7. We start from the Hair et al. (2014) recommendation for the data quality in CB-SEM, i.e., internal assessment, convergent validity, collinearity, and discriminant validity. Finally, the goodness of fit index in the confirmatory factor analysis is presented before the hypothesis test. We have added table 4 to validate the data by the Fornell-Larcker criteria further. These steps are expected to ensure our model exhibits the data's robustness.

We wish these changes could meet your stated comments. We hope you are in good health.

Kind regards,

Authors

Reviewer 4 Report

Topic interesting but in my opinion:

1. The research methodology should be more developed

2. Too many hypotheses are presented

3. Conclusions are too short.

Author Response

Authors' Response: We thank you for your valuable comments. We have:

  1. rewritten methodological section, specifically in subsection 2.2 to provide step-to-step quality tests before conducting the study. This steps are then followed by specific step-to-step presentation in the result section.
  2. The hypotheses present as this article possessed two mediation variables; thus, the indirect paths are also accommodated.
  3. We have added one paragraph in the conclusion section to briefly present our finding.

We wish these changes could meet your stated comments. We hope you are in good health.

Kind regards,

Authors
